# Salivary Biomarkers and Oral Microbial Load in Relation to the Dental Status of Adults with Cystic Fibrosis

**DOI:** 10.3390/microorganisms7120692

**Published:** 2019-12-13

**Authors:** Tamara Pawlaczyk-Kamieńska, Maria Borysewicz-Lewicka, Halina Batura-Gabryel

**Affiliations:** 1Department of Pediatric Dentistry, Poznan University of Medical Sciences, Bukowska 70, 60-812 Poznań, Poland; klstomdz@ump.edu.pl; 2Department of Pulmonology, Allergology and Respiratory Oncology, Poznan University of Medical Sciences, Szamarzewskiego 82/84, 60-569 Poznań, Poland; pulmo@ump.edu.pl

**Keywords:** salivary biomarkers, oral microbiota, oral health status, cystic fibrosis

## Abstract

The mutation of cystic fibrosis transmembrane conductance regulator (CFTR) can modify the physical and chemical properties of saliva, which in turn can affect the oral microflora and oral health in patients with cystic fibrosis (CF). The aim of the study was to examine oral health status, salivary properties, and total oral bacteria count in CF adults. Dental status was assessed using the decayed missing filled surfaces (DMF-S) index, and oral clearness using the approximal plaque index (API). The Saliva-Check BUFFER test was used to assess saliva, and real-time polymerase chain reaction (PCR) test to determine the total oral bacteria count. CF patients in comparison to healthy controls showed a higher level of examined clinical indices, higher total oral bacteria count, lower salivary flow rate, lower salivary pH, and increased viscosity. Conclusions: In CF patients, saliva properties, accompanied by insufficient dental care, might be an essential dental caries risk factor. In CF patients, among the etiological factors for dental caries, the bacterial agent seems to be less significant. The frequent and long-term infectious pharmacotherapy can probably explain that. A great deal of the information collected on the oral environment in CF patients, which has helped us understand the etiological conditions for inflammation and infection in this area of the body, indicates that proper dental care can mostly counteract these pathologies.

## 1. Introduction

The natural environment of the oral cavity and the most important factor affecting the maintenance of oral homoeostasis is saliva. Saliva coats and lubricates tissues, protects teeth surfaces, and mucous membranes against biological, mechanical, and chemical factors [1,2]. It is also the environment of the growth and development of numerous oral microorganisms. Quantity and optimal composition of saliva condition physical and chemical processes, thus contributing to the maintenance of ecological equilibrium. The interaction between the oral ecosystem and the resistance of host tissues plays a significant role in the etiology of oral diseases [1]. Changes in the microbiological composition may increase the pathogenicity of the oral microbiome.

Saliva is a mixture of secretions of minor salivary glands of the lips, palate, tongue, and cheeks and major glands including the parotid, submandibular, and sublingual glands. These are exocrine serous and/or mucous glands with a system of excretory ducts [1,2,3]. The secretion of saliva is a two-stage process. First, acinar cells secrete primary saliva, which is isotonic to plasma. Then, as the fluid travels along the excretory ducts (intercalated and striated), the ion composition of the saliva is significantly modified: Na^+^ and Cl^−^ are reabsorbed, and HCO_3_^−^ and K^+^ are secreted. The electrolyte changing creates the final hypotonic saliva [1,2]. The protein cystic fibrosis transmembrane conductance regulator (CFTR), a chloride pathway dependent on cyclic adenosine-3′5′-monophosphate (cyclic AMP) regulating the transport of Na^+^, K^+^, Cl^−^, and HCO_3_^−^ and water, is an element required for the proper functioning of striated duct cells [2,3,4,5]. The CFTR protein is located on the luminal side of the cell membrane of acini cells and the apical part of cylindrical cells of the striated ducts of the salivary glands [2,6,7,8,9]. Depending on the class of CFTR gene mutation, the synthesis of the CFTR protein may be damage of different grades, which results in chloride pathway malfunctions [4,7,10]. In patients with cystic fibrosis (CF), the CFTR mutation causes the disruption of Na^+^, K^+^, Cl^−^, HCO_3_^−^, and water transport in exocrine ducts of the salivary glands. Consequences may include a modification of the physical and chemical properties of saliva [4,8,11], and the qualitative and quantitative composition of microflora inhabiting the oral cavity, and its health condition.

The first reports concerning irregularities in the salivary glands of CF patients emerged in the 1960s [12,13]. At that time, researchers described changes in the morphological structure concerning the enlargement of glands, the extension of acini, thinning of the epithelium, and the presence of saliva with increased viscosity. The contemporary results indicated a higher concentration of Cl^−^ and Na^+^ in the saliva of CF patients as compared with the control group [14,15], but regarding other salivary parameters, the published data were inconsistent [15,16,17,18,19]. However, the publications referred to concerned children [16,17,19], and just a few included people in their developmental age and adults [14,18,20]. Through the development of CF diagnostic methods and modern CF therapy, the average lifespan of those patients has been considerably prolonged [21], which now allows the gathering of further observations.

Saliva can be useful for diagnosis and monitoring of oral and systemic diseases. As a diagnostic tool, saliva has several advantages, for example, its collection is non-invasive and undemanding, and is a cost-effective method. Saliva can be collected under resting or stimulated conditions, as whole or individual gland saliva. To obtain single gland fluid, usually, there is a necessity of duct cauterization, which is uncomfortable for patients. The collection of the whole saliva may be done by draining, spitting, suction, or the swab method [1,2].

The aims of the studies conducted among CF adults were as follows:Analyze selected physical and chemical properties of saliva and determination of the total count of oral bacteriaAnalyze a survey concerning CF patients′ oral hygiene and use of dental careExamine whether the survey, laboratory, and clinical results are statistically significant and related

## 2. Materials and Methods

The research was conducted in accordance with ethical principles including the World Medical Association Declaration of Helsinki, and was approved by the Ethics Committee of the Poznań University of Medical Sciences, Poland (no. 427/16; Approval date: 6 April 2016). The study involved patients ≥18 years of age with a diagnosis of cystic fibrosis as confirmed by the positive results in sweat and genetic tests. The exclusion criteria included pregnancy or lactation, a systemic disease other than CF, and acute respiratory infections. The control group consisted of generally healthy people who were sex and age compatible with the patients with CF. The exclusion criteria of the healthy group included that of the CF group and smoking, drug use, or alcohol abuse and medicine intake during the examination period. All subjects who participated in the study had the aims and procedures explained to them. Then, they gave informed consent to participate in this study. Finally, 22 patients with CF and 22 people from the control group were selected for the study. 

In each group, women constituted 63.64%. The average age of the patients was 29.14 ± 6.63 years (Figure 1).

The oral health status examination was conducted in accordance with the World Health Organization criteria for epidemiological surveys [22]. Patients were examined by two professionals under artificial light, using a dental mirror. Before the clinical examination, the examiners were calibrated, and Cohen′s kappa value of 0.85 determined an interexaminer agreement.

The clinical examination consisted of the assessment of dental caries occurrence. According to the index used, the number of tooth surfaces with active primary or secondary caries, the number of filled tooth surfaces and teeth extracted due to caries complications were noted. Based on the data obtained during the clinical examination, the caries prevalence was calculated and expressed using the decayed missing filled surfaces (DMF-S) index. The occurrence of the dental plaque was determined with the use of the approximal plaque index (API) by Lange et al. (1977) [23]. Modified API (without the use of a disclosing agent) was used to determine the tooth surface covered with plaque. The assessment criterion was the presence (+) or absence (−) of plaque in interproximal areas. In each quadrant, the occurrence of the plaque was assessed on one side only (i.e., in quadrants 1 and 3 on the oral cavity side and in quadrants 2 and 4 on the vestibule side). The API value was calculated by dividing the number of interproximal areas with plaque by the sum of all surfaces examined. The plaque incidence was expressed as an exact percentage. The scoring criteria were as follows: <25% meant optimal oral hygiene, 25%–39% relatively good, 40%–69% average, and 70%–100% poor hygiene.

A saliva test was performed using the Saliva-Check BUFFER kit (GC, Europe, Poland). The secretion was collected in the morning (08:00–10:00). The patients were asked not to smoke, eat or drink anything, not to brush their teeth, nor to use mouthwash for at least two hours before the test. Their diet on the day preceding the examination should not be different than the usual one. During the examination, the patient was placed in a comfortable sitting position, with mouth and face movements limited to the minimum. On the first step, based on the visual assessment of the appearance of saliva drops at the orifices of the minor glands of the lower lip, the labial secretion was assessed. An appearance of droplets of saliva after more than 60 s was considered a low (below normal) resting flow rate, and under that time, the resting flow rate was considered normal. Then, the resting salivary consistency in the oral cavity was visually assessed. Sticky frothy saliva or frothy bubbly saliva indicates increased viscosity, whereas clear watery saliva indicates normal viscosity. The next step was testing salivary pH. A pH test paper was placed in a vessel with the collected resting saliva for 10 s, the pH value was checked after 1 min, and the coloration of the paper was compared with the scale included in the test. The salivary pH was considered normal if the pH value was 6.8–7.8, moderately acidic if the value was 6.0–6.6, and highly acidic if the value was 5.0–5.8. The quantitative testing of the stimulated saliva (sialometry) was performed after 5 min of chewing of a piece of wax. A volume <3.5 mL indicated very low, 3.5–5 mL low, and >5 mL normal quantity of stimulated saliva. Based on the data obtained, the stimulated saliva flow rate in 1 min was calculated. A salivary flow rate <0.7 mL/minute was determined as very low, 0.7–0.99 mL/minute as low, and ≥1.0 mL/minute as normal [24]. Then, using the point scale of the color indicator of the test paper, the buffering capacity of the stimulated saliva was tested and specified as normal (10–12 points), low (6–9 points) or very low (0–5 point).

The total count of oral bacteria was estimated using a real-time polymerase chain reaction (PCR) test. Diagnostic kit PET Test^®^ plus (MIP Pharma) was used to perform the analysis. Samples were taken from the buccal site of gingival sulcus of four teeth, one in each quadrant: first upper right molar (16), upper right central incisor (11), lower left molar (36), and lower left central incisor (31). Before collecting the sample, the supragingival bacterial plaque was removed, and then the examined area was dried and isolated from the access of saliva with sterile swabs. A sterile paper point included in the kit was introduced using sterilized tweezers to each gingival sulcus for 20 s. Then, each of the four samples was loaded into one testing tube, placed in a transportation set, and shipped. Secured samples were sent to the MIP International Pharma Research GmbH analytical laboratory in Germany.

The CF patients were additionally subjected to a survey using an original questionnaire containing five questions. One question concerned the frequency of oral hygienic procedures performed by the patients, and the rest concerned dentist care: the frequency/regularity of visits to a dentist, reasons for seeing a dentist, date of the last visit, and selection method of a dentist (has it been the same dentist for many years, or if the patient sees different dentists).

Statistical analysis was performed using Statistica program version 12 (StatSoft, Inc., Tulsa, Oklahoma, OK, USA). For statistical inference of the collected data, mean values with standard deviation, median, minimal and maximal values, and percentages were adopted. Statistical differences between CF patients and control group were determined with the Mann–Whitney U test for quantitative variables and the χ^2^ test or Fisher′s exact test for qualitative variables. The Spearman′s rank correlation coefficient was used to determine the correlation between the received values of clinical and salivary indices, and between clinical or salivary indices and the total bacteria count. A value of 0 (*r* = 0) indicates no correlation, *r* ≤ 0.3 is vague correlation, 0.3 < *r* ≤ 0.5 is medium correlation, and r > 0.5 is strong correlation. The level of significance was set at 0.05.

## 3. Results

The results of the clinical examination and salivary tests are presented in Table 1 and Figure 2. The statistical analysis indicated that the CF patients, in comparison with the healthy people, had a significantly higher value of the DMF-S index, a higher number of tooth surfaces with active caries, and a higher number of teeth extracted due to caries. A significant difference was also noted in the percent of the tooth surfaces covered with dental bacterial plaque. In the CF patients, in comparison with the healthy people, significant differences were noted within all unstimulated saliva parameters: lower flow rate, increased salivary viscosity, and lower pH. The average pH of the unstimulated mixed saliva in the CF patients indicated a medium acidic pH, and a normal pH in healthy people. Out of the two analyzed parameters of stimulated saliva, a significant difference was noted in the quantity of saliva/flow rate. Sialometry indicated a lower volume/flow rate of stimulated saliva in the CF patients in comparison with healthy people. No significant difference was noted with regard to the buffer capacity, where both groups were determined as a medium. This analysis was performed in the 21 patients with CF because in one patient, the amount of saliva collected was insufficient. The total bacterial count in the analyzed samples from the CF patients amounted on average to 1.36 × 10^7^ ± 3.8 × 10^7^ and was significantly lower in comparison with the number of microbes in the samples from the control group.

In the CF patients, a negative medium correlation between the stimulated salivary flow rate (volume) and API value, and a strong positive correlation between the buffer capacity of stimulated saliva and the salivary flow rate (volume) were noted (Figure 3). By contrast, in the healthy people, there was a strong negative correlation between salivary pH and the number of tooth surfaces with active dental caries and a strong negative relationship between salivary pH and the total count of bacteria. Such associations were not observed in the CF patients. However, in both groups, a positive correlation between salivary pH and buffering capacity was noted. In the CF patients, the correlation was medium, and in the control group strong.

The surveys on health-promoting behaviors were completed by 16 CF patients (73%). A total of 44% of respondents brushed their teeth twice a day, 31% three times a day, and 25% once a day. During the last three years, 31% of the respondents saw a dentist regularly every six months, 25% once a year, 31% saw a dentist only if they had a toothache, and 13% did not see a dentist at all during that time. The most frequent reason for seeing a dentist as given by the respondents was ache (56%).

## 4. Discussion

The oral cavity is a vast and very diverse ecosystem. The conditions on the lips, cheeks, palate, tongue, periodontal tissues, or teeth are so various that bacteria with extremely different requirements can easily find their optimal habitats there. Aerobic bacteria mainly colonize teeth, but cannot be found in periodontal pockets where almost no oxygen is available. Under the conditions of physiological equilibrium, many of the microbes inhabiting the oral cavity pose no threat due to the defense mechanisms of the host that reduces the number and pathogenicity of microorganisms [25]. The primary defense mechanism regulating the homoeostasis of the oral cavity is saliva, its quantity, composition, and consistency. The physical and chemical properties of saliva, and the quantitative and qualitative structure of oral microflora depend on many factors such as genetic and environmental predispositions (e.g., general condition of the patient, age, diet, oral hygiene, dental and periodontal status, diet including the composition and consistency of food) and pharmacotherapy, especially (as it is in CF patients) if it is long-term or if it is in an inhaled form.

The availability of saliva and the non-invasive method of collecting are essential advantages of this diagnostic material. Saliva contains a large number of inorganic and organic compounds, which can act as oral or systemic diseases biomarkers, a biological indicator of pathologic processes or responses to therapeutic intervention [26]. Studies have proven that the secretion rate, consistency, viscosity, pH level, and buffer capacity are useful indices in defining the risk of the oral diseases [1]. Observations of these parameters significantly enrich the results of the intraoral examination.

The available literature describing the properties of the saliva of CF patients exclusively concern either children and adolescents [16,17,19], or groups including both children and adults [14,18,20]. In the published results of research including adults, the age of CF patients fell within the range of 4–34: in Livnat et al. [20], it was 13.9 ± 7.1 years (six to 23 years); in Gonçavales et al. [14], 12.38 ± 7.1 years (four to 34 years); and in Alkhateeb et al. [18], 11.9 ± 4.0 years (six to 20 years). This age range is lower than that in the present study, where the average age of the participants was 29.14 ± 6.63 years and fell within the range of 20–43. Due to the age difference, the duration of the disease, type, and period of therapy applied, and dietary and hygienic habits were different. Comparing the groups including solely or inclusively children and adults (who constituted this study group), is not entirely satisfactory.

The available publications [16,17,20] showed no significant difference in unstimulated salivary flow rate between CF patients and healthy people. In contrast in our study, the resting flow rate was significantly lower in the CF patients in comparison with healthy people. Our results indicated a significantly lower volume/flow rate of stimulated saliva in the CF patients in comparison with the results in the healthy people, confirming the reports of other authors [14,19]. Moreover, in the CF patients, the stimulated salivary flow rate was lower by 39.45% when compared with the results in healthy people. This finding was consistent with the data provided by da Silva Modesto et al. [19], whose study of the salivary flow rate in CF patients was lower by 36%.

Regarding salivary pH in CF patients, the published reports are inconsistent. Gonçavales et al. [14], who analyzed the pH of mixed stimulated saliva, and Peker et al. [17], who assessed the pH of unstimulated saliva, indicated significantly lower pH in CF patients when compared with the control group. In contrast, Livnat et al. [20] and Peker et al. [16], who examined unstimulated salivary pH, and da Silva Modesto et al. [19], who assessed the pH of stimulated saliva, did not indicate a significant difference between the examined groups. All researchers, except for Peker et al. [16] who used a Saliva-Check BUFFER kit (GC, Europe), used a pH meter in their assessments. In this study, similar to Peker et al. [16], we also used a Saliva-Check BUFFER kit (GC, Europe). The results of our study indicated a significantly lower pH of unstimulated saliva in the CF patients when compared to the healthy controls. Peker et al. [16] and da Silva Modesto et al. [19] did not indicate a significant difference in the buffer capacity of stimulated saliva between the CF patients and the control group. The results of stimulated saliva by Peker at al. [16] and unstimulated saliva by Alkhateeb et al. [18] indicated that the buffer capacity in CF patients was at a medium level. Moreover, Alkhateeb et al. [18] showed that in 26% of CF patients, the buffer capacity was determined as high, in 26% as medium, and 48% as low. The results of our study confirmed the reports above-mentioned, and in both the CF patients and the control group, the buffer capacity was defined as medium.

In summary, in the CF patients, a significantly lower resting and stimulated salivary flow rate, increased viscosity, and lower unstimulated salivary pH were demonstrated. Such physical and chemical properties of saliva may be considered as dental caries risk factors [1].

Hyposalivation is mainly observed in people with chronic diseases (rheumatoid arthritis, diabetes, hypertension, and HIV), and it generally occurs from the impairment of salivary glands, but may also be a side effect of therapies applied (e.g., medications or radiotherapy of the head and neck) [1,24,27,28,29]. It is frequently observed in older people and is usually not associated with individual age, but solely with the overall health condition and medications used [29]. In CF patients, hyposalivation and increased salivary viscosity may be a result of the CFTR mutation, and as a consequence of the improper mechanism of water transport through cell membranes in the excretory ducts of the salivary glands [2,4,8,11], although it may also be a side effect of the pharmacological therapy applied. Moreover, the defective regulation of the exchange of Cl^−^/HCO_3_^−^ in the excretory ducts (the consequence of which is the lack of HCO_3_^−^ secretion to the final saliva), may be the cause of lower salivary pH in CF patients in comparison with the pH of the saliva of healthy people [2]. In our study, we did not determine the concentration of HCO_3_^−^, but we estimated the buffer capacity, which is mainly related to its content. Through the binding of the H^+^, and buffering acids from food or produced by caries inducing bacteria, hydrogen carbonate ions are responsible for maintaining the acid–base equilibrium of saliva. Thus, salivary buffering ability is one of the main factors protecting the oral environment from the drop in pH [30]. In unstimulated saliva, the HCO_3_^−^ buffer is not effective; however, with the saliva secretion stimulation, its concentration increases, and as a consequence, the buffer ability also increases. In our study, we tested the pH of unstimulated saliva, and its low value was likely the effect of the low concentration of HCO_3_^−^, which increased after the stimulation of secretion. The statistical analysis indicated a strong positive correlation in the CF patients between the buffer capacity and the salivary flow rate. Moreover, in both groups, a positive correlation between the unstimulated salivary pH and stimulated salivary buffer capacity was indicated. In the CF patients, the association was medium, and in the healthy controls it was strong, which shows the medium or strong relationship between the salivary pH and buffer capacity, respectively.

In the examined CF patients, who had a significantly lower resting and stimulated salivary flow rate, lower resting pH, and higher viscosity of unstimulated saliva in comparison with the salivary parameters obtained in the control group, a significantly higher dental caries prevalence (DMF-S) was indicated, which confirms the influence of saliva on caries development and progression. However, the results do not confirm the reports of Alkhateeb et al. [18], who noted a negative correlation between salivary pH and dental caries prevalence in CF patients. We observed such an association in the healthy controls, where the statistical analysis additionally indicated a negative relationship between salivary pH and the total bacteria count. Gonçavales et al. [31], who used an animal model and the diet applied, has proven a significant increase in dental caries intensity and the number of caries inducing bacteria, after an application in CF animals for a period of five weeks of a feed rich in carbohydrates. The analysis of mixed stimulated saliva collected from them indicated a significant decrease in HCO_3_^−^ ion concentration and salivary pH in comparison with the initial levels. The authors suggested that a low salivary pH in CF mice resulted from a low concentration of HCO_3_^−^ and their abnormal secretion to the final saliva. Such condition promotes the adhesion and development of acidophilic microbes and thus may be a factor increasing the risk of dental caries.

Another oral diseases factor is long-lasting dental bacterial plaque [25]. In the CF patients, a negative correlation between the salivary flow rate and the percentage of teeth covered with bacterial plaque (API value) was noted. These results confirm the reports concerning older people [28] or patients with systemic diseases [27], in which besides lower saliva flow rate, a higher aggregation of plaque was also observed. This finding may suggest that in CF patients, besides the reduced salivary flow rate, an additional factor promoting the accumulation of dental plaque and reduction of oral clearance level may be a necessity for a consumption high energy, rich in carbohydrates diet. The ability to form bacterial dental plaque is a biological process that is undoubtedly affected by individual and external factors such as patient′s oral habits: frequency and accuracy of hygienic routines, and dietary habits. In CF patients, in comparison with healthy people, despite a higher accumulation of bacterial plaque, the total number of microbes was significantly lower, which can probably be explained by broad-spectrum antibiotic therapy (also by inhalation).

The clinical examination showed that CF patients demonstrated lesser oral health care than healthy people, which was reflected in a significantly higher number of teeth surfaces with active caries, a higher number of extracted teeth, and increased quantity of dental bacterial plaque. Moreover, the survey conducted among CF patients indicated that the main reason for seeing a dentist was a toothache or visible carious defect, which suggests that the treatment is rather interventional. Only a few of them control their oral health regularly. This fact and the high accumulation of dental plaque in CF patients are evidence of the lack of proper dental care and low level of oral hygiene. Meanwhile, the observation from bacteriological studies is puzzling. The total oral bacteria count was significantly lower in the CF patients in comparison with the healthy people, which may suggest lower pathogenicity of such deposits, both regarding dental caries and periodontal diseases [32,33].

Providing CF patients with permanent dentist care and establishing particular prophylactic guidelines for this group of patients are significant factors allowing the maintenance of a healthy oral cavity. Proper prophylactics and the need to adapt oral health-promoting recommendations by patients such as removing bacterial dental deposits, improving saliva features, and the application of preparations increasing enamel resistance to pathogens give a chance to maintain dentition for throughout one’s entire life and eliminate the potential pathogenic lesions within the oral cavity. A large aggregation of bacterial plaque may be a reservoir of caries and periodontal pathogens. In CF patients, in comparison with healthy people, the adverse salivary properties such as reduced salivary flow rate, increased viscosity, and lower pH do not guarantee oral protection against microbes. Insufficient dental care in CF patients is demonstrated not only in poor oral hygiene, but also cannot reduce carious outbreaks and caries complications. This condition leads to the loss of dental hard tissues and as a consequence, the loss of teeth. Dental guidelines for CF patients, reflecting the prophylactics and treatment needs of CF children and young adults, could be a pathway for each dentist and could reduce the risk of oral diseases. The phrase “large accumulation of dental bacterial plaque” is a generalization that proves the deficit of the natural mechanism of microorganism removal from the oral cavity (saliva) and through hygienic procedures. However, the pathogenicity of the dental plaque is not related to its quantity, but its quality (composition) and depends on the profile of the microbes. It is conditioned not only by individual features of saliva, but also by the diet applied (consistency and composition), the oral hygiene preparations used, and frequency and accuracy of the tooth brushing. The “poor condition of oral hygiene” confirmed by the study, which shall be understood as plaque deposits on the enamel surface, and their growth in thickness, does not provide information on the pathogenicity. As we have proven in our earlier study, despite the presence in CF patients of large deposits of bacterial plaque that are a periodontal risk factor, the clinical indicators of periodontal diseases did not indicate the presence of an acute inflammation [32].

To the best of our knowledge, this is the first study on oral health solely covering CF adults. However, in this publication, the conclusions may be restricted by the number of CF patients. Such a number did not allow for the analysis of the results taking into account the class of CFTR mutation, which is likely to affect the physical and chemical properties of saliva significantly. One should hope that the development of therapies and care for CF patients allow further observations in more numerous groups of CF patients and contribute to the prolongation of their ontogenetic life.

## 5. Conclusions

(1)In CF patients, saliva properties, accompanied by insufficient dental care, might be an essential etiological factor in the development and progression of dental caries.(2)Among the etiological factors for dental caries in CF patients, the bacterial agent seems to be less significant. The frequent and long-term infectious pharmacotherapy can probably explain that.(3)A great deal of the information collected on the oral environment in CF patients, which has helped us understand the etiological conditions for inflammation and infection in this area of the body, indicating that proper dental care can mostly counteract these pathologies.

## Figures and Tables

**Figure 1 microorganisms-07-00692-f001:**
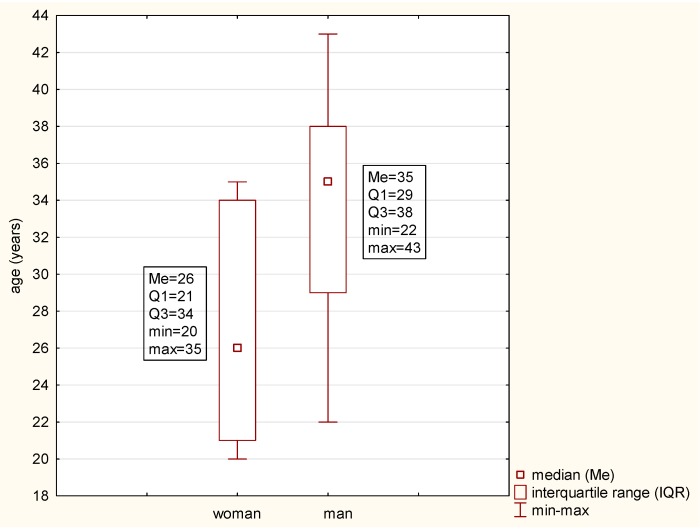
Age distribution in the studied subjects.

**Figure 2 microorganisms-07-00692-f002:**
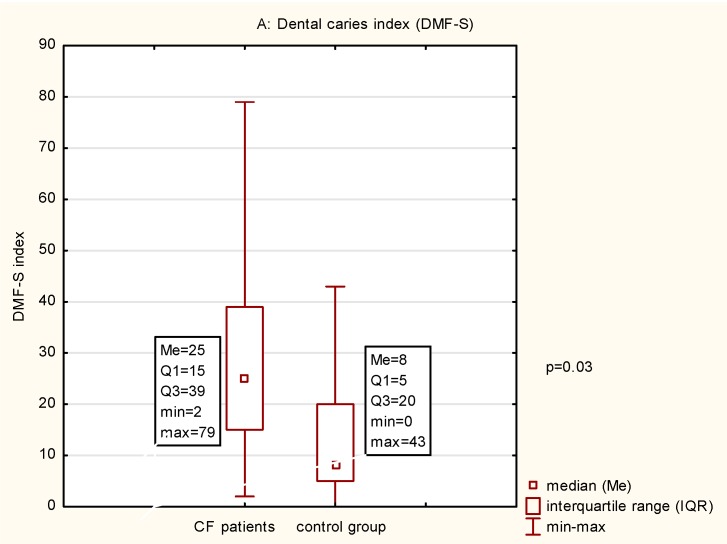
(**A**–**F**) Data comparison graphs of the results of Mann-Whitney tests between cystic fibrosis patients and healthy controls. Abbreviations: CF, cystic fibrosis; DMF-S, decayed missing filled surfaces; API, approximal plaque index

**Figure 3 microorganisms-07-00692-f003:**
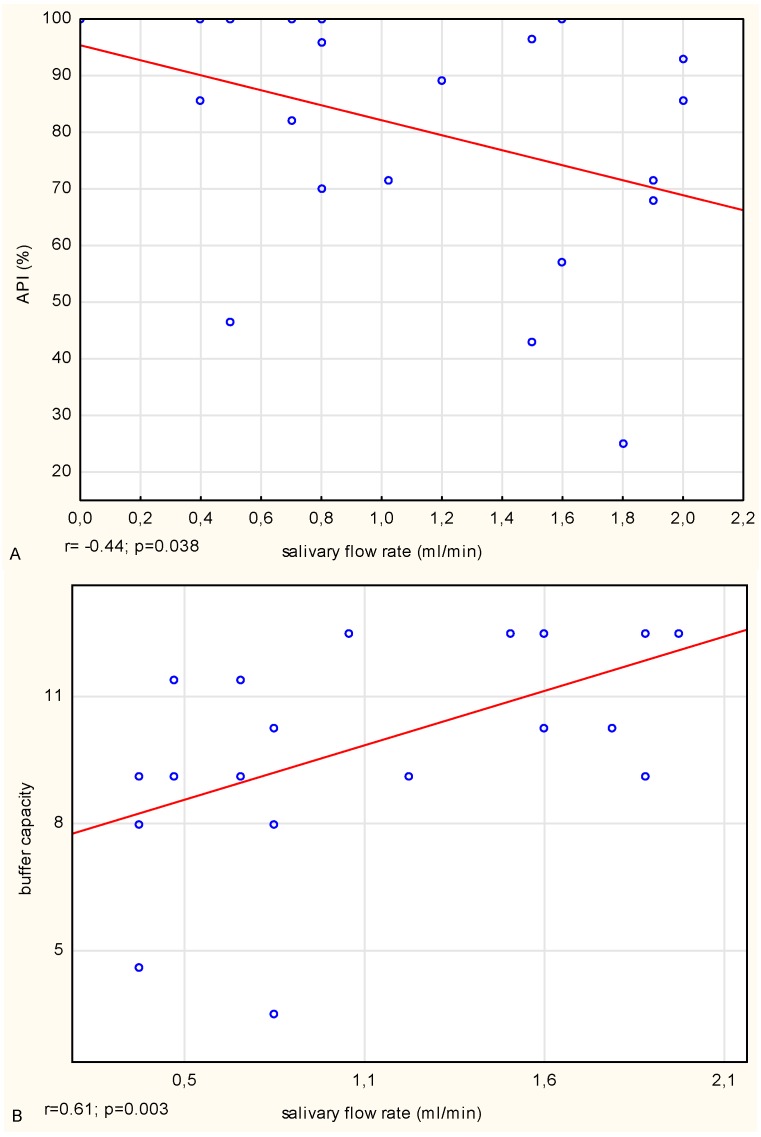
(**A**,**B**) Spearman correlation analysis between the approximal plaque index (API) and salivary flow rate, and between the buffer capacity of stimulated whole saliva and salivary flow rate in cystic fibrosis (CF) patients.

**Table 1 microorganisms-07-00692-t001:** Oral health status and salivary parameters in cystic fibrosis patients and healthy controls.

			CF Patients *n* (%)	Control Group *n* (%)	*p*-Value
D-S > 0			19 (86.36)	6 (27.27)	0.0001
M-S > 0			14 (63.63)	7 (31.82)	0.17
F-S > 0			20 (90.91)	19 (86.36)	0.65
API	<25%		0 (0.00)	1 (4.55)	0.34
	25–39%		1 (4.54)	6 (27.27)	0.03
	40–70%		5 (22.73)	5 (22.73)	0.98
	>70%		16 (72.73)	10 (45.45)	0.07
unstimulated saliva	flow rate	Normal	11 (50.00)	22 (100)	0.0001
		Low	11 (50.00)	0 (0.00)	0.0001
	consistency	normal viscosity	3 (13.64)	22 (100)	<0.0001
		increased viscosity	19 (86.36)	0 (0.00)	<0.0001
	pH	Normal	9 (40.91)	16 (72.73)	0.04
		moderately acidic	11 (50.00)	5 (22.73)	0.06
		highly acidic	2 (9.09)	1 (4.54)	0.50
stimulated saliva	volume/flow rate	Normal	11 (50.00)	22 (100)	0.0001
		Low	5 (22.73)	0 (0.00)	0.02
		very low	6 (27.27)	0 (0.00)	0.009
	buffering capacity	Normal	12 (57.14) *	9 (40.91)	0.29
		Low	7 (33.33) *	13 (59.09)	0.08
		very low	2 (9.52) *	0 (0.00)	0.15

* buffering capacity was tested in 21 CF patient. Abbreviations: CF, cystic fibrosis; DMF-S, decayed missing filled surfaces; D-S, decayed surfaces; M-S, missing surfaces; F-S, filled surfaces; API, approximal plaque index.

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
