# Peer review of "Salivary Biomarkers and Oral Microbial Load in Relation to the Dental Status of Adults with Cystic Fibrosis"

_microorganisms, 2019, doi:10.3390/microorganisms7120692_

Round 1

Reviewer 1 Report

The manuscript is clearly written and the results are well presented.

I would like to suggest authors enhance the introduction, by the giving small (2-3 sentences) off common approaches used for Salivary Biomarkers.

The materials and methods section should be improved by better explaining the use of the Statistica program software.

I would suggest the inclusion of the following works to increase the bibliographic:

1) Atomic force microscopy of bacteria from periodontal subgingival biofilm: Preliminary study results  DOI: 10.4103/1305-7456.110155

2) Molecular biomarkers related to oral carcinoma: Clinical trial outcome evaluation in a literature review DOI: 10.1155/2019/8040361

Author Response

We would like to thank the reviewer for the valuable comments and suggestions, which help to improve the quality of this manuscript. Correction of grammatical errors and English improvement were carried done by the professional editing service as suggested. We have also added Figure 3 (Spearman correlation analysis between approximal plaque index (API) and salivary flow rate, and between buffer capacity of stimulated whole saliva and salivary flow rate in cystic fibrosis (CF) patients ) which unintentionally we had not attached in the previous version. We apologize for this mistake. 

The following are our point-by-point responses (the reviewer’s comments are in italics):

The manuscript is clearly written and the results are well presented.

Response:

We appreciate the positive feedback from the reviewer. As suggested by the reviewer, we have reviewed carefully the manuscript.

I would like to suggest authors enhance the introduction, by the giving small (2-3 sentences) off common approaches used for Salivary Biomarkers.

Response:

Thank you very much for this comment. We have rewritten the introduction section, and we have added some sentences to the manuscript:

PAGE 2 LINE 68: Saliva can be useful for diagnosis and monitoring of oral and systemic diseases. As a diagnostic tool, saliva has several advantages, e.g. its collection is non-invasive and undemanding; it is a cost-effective method. Saliva can be collected under resting or stimulated conditions; as whole or individual gland saliva. To obtain single gland fluid, usually, there is necessity duct cauterization, which is uncomfortable for patients. The collection of the whole saliva may be done by draining, spitting, suction or the swab method [1, 2].

The materials and methods section should be improved by better explaining the use of the Statistica program software.

Response:

Thank you very much for pointing out this problem. We have rewritten this part of the Material and Methods section as:

PAGE 4 LINE 148: A statistical analysis was performed using Statistica program version 12 (StatSoft, Inc., Tulsa, United States). For statistical inference of the collected data, mean values with standard deviation , median, minimal and maximal values, and percentages were adopted. Statistical differences between CF patients and control group were determined with the Mann–Whitney U test for quantitative variables and the χ2 test or Fisher’s exact test for qualitative variables. The Spearman's rank correlation coefficient was used to determine the correlation between the received values of clinical and salivary indices, and between clinical or salivary indices and the total bacteria count. A value of 0 (r = 0) indicates no correlation, r ≤ 0.3 - vague correlation, 0.3 < r £ 0.5 - medium correlation and r > 0.5 - strong correlation. The level of significance was set at 0.05.

I would suggest the inclusion of the following works to increase the bibliographic:

1) Atomic force microscopy of bacteria from periodontal subgingival biofilm: Preliminary study results  DOI: 10.4103/1305-7456.110155

2) Molecular biomarkers related to oral carcinoma: Clinical trial outcome evaluation in a literature review DOI: 10.1155/2019/8040361

Response:

Thank you very much for the very interesting publications and we appreciate the suggestion with regard to the increasing the bibliographic of our article. We have found the papers very interesting and we have decided to cite this publication. Thank you very much for your suggestion. We have also added some sentences regarding saliva in the Discussion section: 

PAGE 8 LINE 218: The availability of saliva and the non-invasive method of collecting it are essential advantages of this diagnostic material. Saliva contains a large number of inorganic and organic compounds, which can act as oral or systemic diseases biomarkers - a biological indicator of pathologic processes or responses to therapeutic intervention [26]. Studies have proven that the secretion rate, consistency, viscosity, pH level and buffer capacity are useful indices in defining the risk of the oral diseases [1].

Reviewer 2 Report

Article is very interesting connection between dentistry and rare diseases. I would like suggest that Authors should write the full name and abbreviation when they appear for the first time in the text, e.g. in the abstract is only "CF", and there should be "cystic fibrosis (CF)", etc. Similarly: DMF-S, API. In Methods real-time PCR test should be more described, including used primers, conditions of reaction. How were taken samples from the gingival crevice? What was used to receiving samples: filter paper, needle? How many microliters of gingival fluid was obtained? In Discussion dates/years after names of authors are unnecessary - please remove these.

Author Response

We sincerely thank the reviewer for constructive criticisms and comments, which were of great help in revising the manuscript. We have revised our manuscript. Correction of grammatical errors and English improvement were carried done by the professional editing service as suggested. We have also added Figure 3 (Spearman correlation analysis between approximal plaque index (API) and salivary flow rate, and between buffer capacity of stimulated whole saliva and salivary flow rate in cystic fibrosis (CF) patients ) which unintentionally we had not attached in the previous version. We apologize for this mistake. 

Please, find below our responses (the reviewer’s comments are in italics).

Article is very interesting connection between dentistry and rare diseases.

Response:

We would like to thank the reviewer for the kind words about our paper.

I would like to suggest that Authors should write the full name and abbreviation when they appear for the first time in the text, e.g. in the abstract is only "CF", and there should be "cystic fibrosis (CF)", etc. Similarly: DMF-S, API.

Response:

Thank you very much for pointing out this problem. Your suggestion is very valuable. We have carefully revised the manuscript and added the full name of used abbreviation to the text.

In Methods real-time PCR test should be more described, including used primers, conditions of reaction. How were taken samples from the gingival crevice? What was used to receiving samples: filter paper, needle? How many microliters of gingival fluid was obtained?

Response:

Thank you very much for this vulnerable comment. We have carefully revised this section of our manuscript and we have added some sentences regarding used PCR test:

PAGE 4 LINE 137: Before collecting the sample, the supragingival bacterial plaque was removed, and then the examined area was dried and isolated from the access of saliva with sterile swabs. A sterile paper point included in the kit was introduced using sterilized tweezers to each gingival sulcus for 20 seconds. Then, each of the four samples was loaded into one testing tube, placed in a transportation set and shipped

In Discussion dates/years after names of authors are unnecessary - please remove these.

Response:

The correction has been made.